# Legendre Deep Neural Network (LDNN) and its application for approximation of nonlinear Volterra–Fredholm–Hammerstein integral equations

## Abstract

Various phenomena in biology, physics, and engineering are modeled by differential equations. These differential equations including partial differential equations and ordinary differential equations can be converted and represented as integral equations. In particular, Volterra–Fredholm–Hammerstein integral equations are the main type of these integral equations and researchers are interested in investigating and solving these equations. In this paper, we propose Legendre Deep Neural Network (LDNN) for solving nonlinear Volterra–Fredholm–Hammerstein integral equations (V-F-H-IEs). LDNN utilizes Legendre orthogonal polynomials as activation functions of the Deep structure. We present how LDNN can be used to solve nonlinear V-F-H-IEs. We show using the Gaussian quadrature collocation method in combination with LDNN results in a novel numerical solution for nonlinear V-F-H-IEs. Several examples are given to verify the performance and accuracy of LDNN.

## 1 Introduction

Deep neural networks are a main and beneficial part of machine learning family which are applied in various areas including speech processing, computer vision, natural language processing and image processing (LeCun et al., 2015; Krizhevsky et al., 2012). Also, the approximation of the functions is a significant branch in scientific computational and achieving success in this area is considered by some research (Tang et al., 2019; Hanin, 2019). Solving differential equations is the other main branch of scientific computational which neural networks and deep learning have been shown success in this area. (Lample & Charton, 2019; Berg & Nyström, 2018; Raissi et al., 2019). Various phenomena in biology, physics, finance, neuroscience and engineering are modeled by differential equations (Courant & Hilbert, 2008; Davis, 1961). In recent years, several researchers studied the solving differential equations via deep learning or neural networks. differential equations consists of ordinary differential equations, partial differential equations and integral equations. (Sirignano & Spiliopoulos, 2018; Lu et al., 2019; Meng et al., 2020). It is notable that the various numerical methods are applied for solving differential equations. Homotopy analysis method (HAM) (Liao, 2012) and variational iteration method (VIM) (He & Wu, 2007) are known as analytical/semi-analytical methods. Usually, spectral methods (Canuto et al., 2012), Runge-Kutta methods (Hairer et al., 2006), the finite difference methods (FDM) (Smith, 1985) and the finite element methods (FEM) (Johnson, 2012) are considered as the popular numerical methods. When the complexity of the model does not allow us to obtain the solution explicitly, numerical methods are a proper selection for finding the approximate solution for the models. Recently, some of the machine learning methods are applied for solving differential equations. Chakraverty & Mall (2017) introduced orthogonal neural networks which used orthogonal polynomials in the structure of the network. Raja et al. (2019) applied meta-heuristic optimization algorithm to neural network for obtaining the solution of differential equations. Moreover, other methods of machine learning such as support vector machine (Vapnik, 2013) are used to approximate the solution of the models. Least squares support vector machines are considered in these researches (Hajimohammadi et al., 2020; Mehrkanoon & Suykens, 2015). Baker et al. (2019) selected deep neural networks for solving the differential equations. Pang et al.

(2019) introduced a new network to find the solution of the different equations. Han et al. (2018) solved high-dimensional problems via deep networks. Also, Long et al. (2018) and Raissi et al. (2019) introduced a group of the equations which solved by deep learning. Furthermore, He et al. (2018) and Molina et al. (2019) investigated the effect of the activation function on networks.

In this paper, we concern nonlinear Volterra–Fredholm–Hammerstein integral equations (V-F-H-IEs) and try to obtain the solution of them via deep neural network. We present a new numerical approach of machine learning which is a combination of deep neural network and Legendre collocation method. This approach is useful for solving the differential equations and we applied it for solving nonlinear V-F-H-IEs. We used Legendre collocation method to our network for perfect the numerical computations and enhancement the performance the network.

## 2    LEGENDRE DEEP NEURAL NETWORK (LDNN)

The main purpose of introducing LDNN is to apply it for solving differential models. Indeed, this purpose is to expand the utilization of deep learning networks in the field of scientific computing, especially the solution of differential equations. Moreover, this network has the advantages of solving equations by deep learning as well as numerical methods such as collocation method used to achieve better solution to the equations. LDNN presents a combination of a deep neural network and Legendre collocation method. In fact, our network consists of two networks which have connected consecutive to each other. The first network is a feed forward neural network which has an orthogonal Legendre layer. The second network includes operation nodes to create the desired computational model. In recent decades, numerical methods especially collocation method are popular methods for solving differential equations. In the collocation method, first an approximation of the solution is expanded by using the sum of the basic functions. The basic functions consists of the orthogonal polynomials such as Legendre polynomials.Then this approximation is placed in the differential equation. By considering the appropriate set of candidate points, an attempt is made to obtain the unknown coefficients of the basic functions so that the solution satisfies the equation in a set of candidate points. The first network is applied to creat the approximation of the solution. This approximation can be known as the scattered data interpolation problem. The second network is used to obtain the desired equation so that the solution satisfies it. The structure of LDNN is described in detail at the following rest.

Consider that the first network has a $\mathcal{M}$-layer which defined as follows:

$$
\begin{aligned}
\mathcal{H}_0 &= \boldsymbol{x}, \quad \boldsymbol{x} \in \mathbb{R}^d, \\
\mathcal{H}_1 &= L(\boldsymbol{W}^{(1)}\mathcal{H}_0 + \boldsymbol{b}^{(1)}), \\
\mathcal{H}_i &= f(\boldsymbol{W}^{(i)}\mathcal{H}_{i-1} + \boldsymbol{b}^{(i)}), \quad 2 \leq i \leq \mathcal{M} - 1, \\
\mathcal{H}_{\mathcal{M}} &= \boldsymbol{W}^{(\mathcal{M})}\mathcal{H}_{\mathcal{M}-1} + \boldsymbol{b}^{(\mathcal{M})}.
\end{aligned}
$$

where $\mathcal{H}_0$ is the input layer with $d$ dimension. $\mathcal{H}_i$, $1 \leq i \leq \mathcal{M} - 1$ are hidden layers, $L = [L_0, L_1, ...L_n]^T$ which $L_i$ are $i$-th degrees of Legendre orthogonal polynomials, $\mathcal{H}_1$ is an orthogonal layer, $f$ is the hyperbolic tangent activation function or other commonly used activation functions. $\boldsymbol{W}^{(i)}$, $i = 1, \cdots, \mathcal{M}$ are the weight parameters and $\boldsymbol{b}^{(i)}$, $1 \leq i \leq \mathcal{M}$ are the bias parameters. $\mathcal{H}_{\mathcal{M}}$ is the output layer. It is notable that the second network is applied to obtain the desired differential model. This aim is possible by using operation nodes including integrals, derivatives, and etc. These nodes are applied to the output of the first network. Moreover, automatic differentiation (AD) (Baydin et al., 2017) and Legendre Gaussian integration (Shen et al., 2011) have been used in network computing to obtain more accurate and fast calculations. How to train the network and set the parameters are also important points. Supervised learning method is used to train network. The cost function for setting parameters is defined as follows:

$$
\text{CostFun} = \min(y_t - y_p) + \min(R_m). \tag{1}
$$

where $y_t$ is an exact value of the model and $y_p$ is a predicted value of the LDNN. The definition of $R_m$ is explained in section 3.The minimization of CostFun is obtained by performing Adam algorithm (Kingma & Ba, 2015) and the L-BFGS method (Liu & Nocedal, 1989) on mean squared errors of training data set.

## 2.1 Legendre Polynomials

Legendre polynomials (Shen et al., 2011) are a main series of orthogonal polynomials which denoted by $L_n(\eta)$, are defined as:

$$L_n(\eta) = \frac{1}{2^n} \sum_{\ell=0}^{\left[\frac{n}{2}\right]} (-1)^\ell \frac{(2n - 2\ell)!}{2^n \ell!(n - \ell)!(n - 2\ell)!} \eta^{n-2\ell} \tag{2}$$

Legendre polynomials are defined in $[-1, 1]$ domain and have the recurrence formula in the following form:

$$(n + 1)L_{n+1}(\eta) = (2n + 1)\eta L_n(\eta) - nL_{n-1}(\eta), \quad n \geq 1,$$
$$L_0(\eta) = 1, \qquad L_1(\eta) = \eta. \tag{3}$$

Orthogonality relation for these polynomials is as follows:

$$\int_{-1}^{1} L_n(\eta)L_m(\eta)\mathrm{d}\eta = \gamma\delta_{n,m}, \tag{4}$$

where $\delta_{n,m}$ is a delta Kronecker function and $\gamma = \frac{2}{2n+1}$.
The weight function of them is $\mathcal{W}(\eta) = 1$. Some following useful properties of Legendre polynomials are defined:

$$L_n(-\eta) = (-1)^n L_n(\eta), \tag{5}$$
$$|L_n(\eta)| \leq 1, \quad \forall \eta \in [-1, 1], \ n \geq 0, \tag{6}$$
$$L_n(\pm 1) = (\pm 1)^n, \tag{7}$$
$$(2n + 1)L_n(\eta) = L'_{n+1}(\eta) - L'_{n-1}(\eta), \quad n \geq 1. \tag{8}$$

## 3 Nonlinear Volterra–Fredholm–Hammerstein integral equations and LDNN

The general form of nonlinear Volterra–Fredholm–Hammerstein integral equations (V-F-H-IEs) is as follows:

$$y(x) = g(x) + \xi_1 \int_0^x K_1(x, s)\varphi_1(s, y(s))\mathrm{d}s + \xi_2 \int_0^1 K_2(x, s)\varphi_2(s, y(s))\mathrm{d}s, \quad x \in [0, 1]. \tag{9}$$

where $\xi_1$, $\xi_2$ are fixed, $g(x)$, $K_1(x, s)$ and $K_2(x, s)$ are given functions and $\varphi_1(s, y(s))$, $\varphi_2(s, y(s))$ are nonlinear functions. The aim is to find the proper $y(x)$. In order to use the LDNN, reformulated Eq. (9) in the following form:

$$R_m = -y(x) + g(x) + \xi_1 \int_0^x K_1(x, s)\varphi_1(s, y(s))\mathrm{d}s + \xi_2 \int_0^1 K_2(x, s)\varphi_2(s, y(s))\mathrm{d}s, \quad x \in [0, 1]. \tag{10}$$

$y(x)$ is approximated by the first network of the LDNN.

$$y(x) \approx \mathcal{H}_\mathcal{M}. \tag{11}$$

Furthermore, we applied Legendre–Gauss integration formula (Shen et al., 2011):

$$\int_{-1}^{1} h(X)\mathrm{d}X = \sum_{j=0}^{N} \omega_j h(X_j) \tag{12}$$

where $\{X_j\}_{j=0}^N$ are the roots of $L_{n+1}$ and $\{\omega_j\}_{j=0}^N = \frac{2}{(1-X_j^2)(L'_{n+1}(X_j))^2}$. Here, we should transfer the $[0, x]$ and $[0, 1]$ domains into the $[-1, 1]$ domain. It is possible by using the following transformation:

$$t_1 = \frac{2}{x}s - 1, \quad t_2 = 2s - 1.$$

Consider

$$Z_1(x,s) = K_1(x,s)\varphi_1(s,y(s)),$$
$$Z_2(x,s) = K_2(x,s)\varphi_2(s,y(s)).$$

we have

$$R_m = -y(x) + g(x) + \xi_1 \frac{x}{2} \int_{-1}^{1} Z_1(x, \frac{x}{2}(t_1+1))\mathrm{d}t_1 + \frac{\xi_2}{2} \int_{-1}^{1} Z_2(x, \frac{x}{2}(t_2+1))\mathrm{d}t_2. \quad (13)$$

by using Legendre–Gauss integration formula, the below form is concluded:

$$R_m = -y(x) + g(x) + \xi_1 \frac{x}{2} \sum_{j=0}^{N_1} \omega_{1j} Z_1(x, \frac{x}{2}(t_{1j}+1)) + \frac{\xi_2}{2} \sum_{j=0}^{N_2} \omega_{2j} Z_2(x, \frac{x}{2}(t_{2j}+1)). \quad (14)$$

The second network of LDNN and its nodes makes $R_m$. The architecture of LDNN for solving nonlinear V-F-H-IEs is represented in Figure 1.

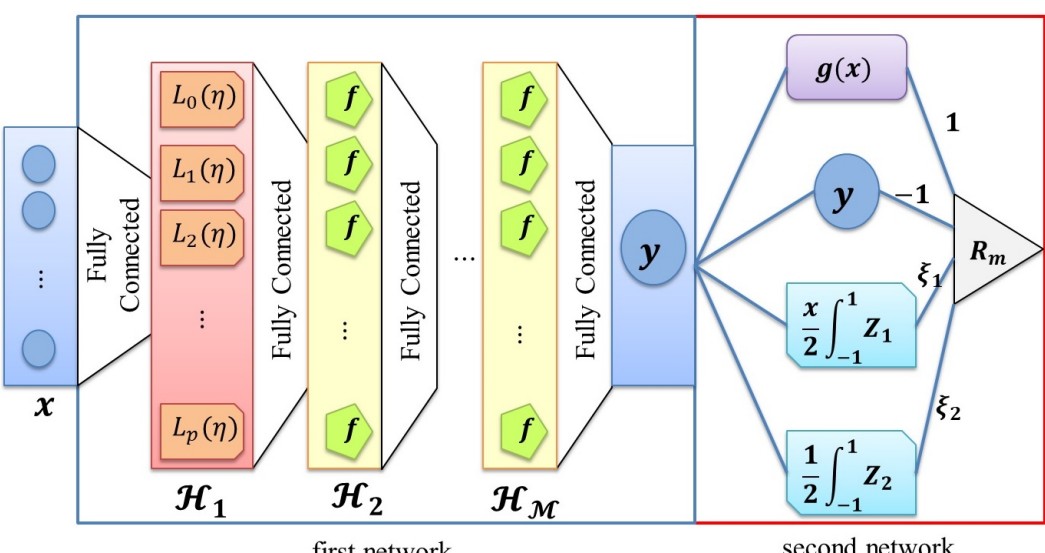

Figure 1: The architecture of LDNN for solving nonlinear V-F-H-IEs. The first network approximates the solution of IE $y(x)$. This network has $\mathcal{M}$-layer and feed forward neural network is the structure of it. $\mathcal{H}_1$ is introduced as a orthogonal layer which consists of $p$ neurons with $\{L_i\}_{i=0}^{p}$ (Legendre polynomials) as activation functions. Other layers have $f$, hyperbolic tangent as activation functions. The second network with the nodes makes the desired model and the output of it, is $R_m$ (consider Eq. (14)). The outputs of LDNN are $y(x)$ and $R_m$.

## 4 NUMERICAL RESULTS

In order to present the accuracy and performance of the LDNN for solving nonlinear V-F-H-IEs and justify the efficiency of the proposed method, several examples are given. The convergence behavior of the LDNN is reported by using the following parameters:
The exact value $y_t$, the predicted value $y_p$ and the absolute error (Error) in some points of test data are reported in various tables. The number of the train data $m_1$, the number of Legendre quadrature points $(N_1, N_2)$, the number of the test data $m_2$, the structure of network $\mathcal{M}$-layers, $L_2^{train}$ and $L_2^{test}$ are shown in Table 1. $L_2^{train}$ and $L_2^{test}$ are calculated as follows:

$$L_2^{train} = ||y_t - y_p||_2 = [\sum_{j=1}^{m_{tr}} (y_t(x_j) - y_p(x_j))^2]^{\frac{1}{2}},$$

$$L_2^{test} = ||y_t - y_p||_2 = [\sum_{j=1}^{m_{te}} (y_t(x_j) - y_p(x_j))^2]^{\frac{1}{2}}, \quad (15)$$

Table 1: The LDNN parameters for all the experiments. The structure of $\mathcal{M}$-Layers indicates by $[d, NL^{(1)}, NL^{(2)}, \cdots, NL^{(\mathcal{M}-1)}, 1]$. This network has $d$ dimension in input layer, $\mathcal{M} - 1$ hidden layers with $NL^{(\ell)}$, $2 \leq \ell \leq \mathcal{M} - 1$, neurons in each layer and one output which approximates the $y(x)$. All the experiments have 4 hidden layers.

| **Experiment** | $\mathcal{M}$-Layers | $m_1$ | $(N_1, N_2)$ | $L_2^{train}$ | $m_2$ | $L_2^{test}$ |
|---|---|---|---|---|---|---|
| Experiment 1 | $[1, 10, 30, 20, 10, 1]$ | 500 | $(50, -)$ | 3.937867e−09 | 100 | 4.015095e−09 |
| Experiment 2 | $[1, 10, 30, 20, 10, 1]$ | 500 | $(50, 50)$ | 7.156029e−09 | 100 | 7.537263e−09 |
| Experiment 3 | $[1, 10, 30, 20, 10, 1]$ | 500 | $(50, 50)$ | 1.347132e−09 | 100 | 1.659349e−08 |
| Experiment 4 | $[1, 10, 30, 20, 10, 1]$ | 500 | $(50, 50)$ | 9.182442e−09 | 100 | 1.107755e−09 |

Table 2: The exact value, the predicted value and the absolute error (Error) in several test points on $[0, 1]$ domain for Experiment 1.

| $x$ | exact value ($y_t = e^x$) | predicted value ($y_p$) | Error |
|---|---|---|---|
| 0.0 | 1.0 | 1.000000049 | 4.90000001e−08 |
| 0.2 | 1.22140276 | 1.221402765 | 4.99999997e−09 |
| 0.4 | 1.4918247 | 1.49182494 | 2.40000000e−07 |
| 0.6 | 1.8221188 | 1.82211831 | 4.90000000e−07 |
| 0.8 | 2.22554093 | 2.225540981 | 5.09999998e−08 |
| 1.0 | 2.71828183 | 2.71828179 | 4.00000002e−08 |

The Tensorflow package of Python version 3.7.0. is applied for writing the code of all experiments. Adam algorithm is stoped when the number of iteration is up to 5000 and L-BFGS method is stoped when it converges. The figures are obtained on the test data set.

## 4.1 EXPERIMENT 1

Suppose that we have the following model (Yousefi & Razzaghi, 2005):

$$y(x) = e^x - \frac{1}{3}e^{3x} + \frac{1}{3} + \int_0^x y^3(s)\mathrm{d}s, \quad x \in [0, 1]. \tag{16}$$

It has the exact solution $y(x) = e^x$. Table 2 represents the exact value, the predicted value and the absolute error (Error) in several test points on $[0, 1]$ domain. 50 points of shifted Legendre quadrature points are applied for training LDNN. The number of train data set is 500 and the number of test data set is 100. Figure 2 shows the illustrated comparison between $y_t$ and $y_p$.

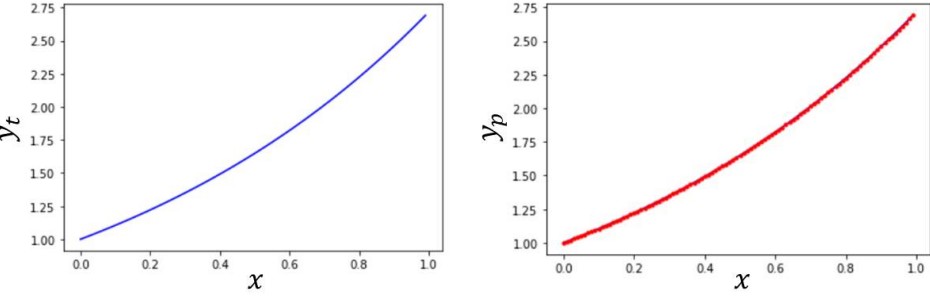

Figure 2: Results of Experiment 1. Exact solution $y_t(x) = e^x$, predicted solution $y_p(x)$ by LDNN.

Table 3: The exact value, the predicted value and the absolute error (Error) in several test points on $[0, 1]$ domain for Experiment 2.

| $x$ | exact value ($y_t = \cos(x)$) | predicted value ($y_p$) | Error |
|-----|------|------|------|
| 0.0 | 1.0 | 1.000000059 | 5.90000000e−08 |
| 0.2 | 0.98006658 | 0.98006683 | 2.50000000e−07 |
| 0.4 | 0.92106099 | 0.92106083 | 1.50000000e−07 |
| 0.6 | 0.82533561 | 0.82533555 | 6.00000000e−08 |
| 0.8 | 0.69670671 | 0.69670670 | 9.99999994e−09 |
| 1.0 | 0.54030231 | 0.54030237 | 6.00000001e−08 |

### 4.2   EXPERIMENT 2

Suppose that we have the following model (Razzaghi & Ordokhani, 2002):

$$y(x) = 1 + \sin^2(x) + \int_0^1 K(x,s)y^2(s)\mathrm{d}s, \quad x \in [0,1]. \tag{17}$$

where

$$K(x,s) = \{ \begin{array}{ll} -3\sin(x-s), & 0 \le s \le x; \\ 0 & x \le s \le 1. \end{array} \tag{18}$$

It has the exact solution $y(x) = \cos(x)$. The exact value, the predicted value and the absolute error (Error) in several test points on $[0,1]$ domain are reported in Table 3. 50 points of shifted Legendre quadrature points are applied for training LDNN. The number of train data set is 500 and the number of test data set is 100. Figure 3 shows the illustrated comparison between $y_t$ and $y_p$.

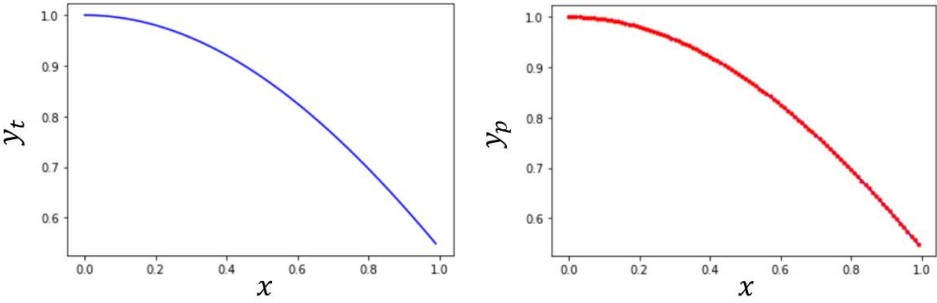

Figure 3: Results of Experiment 2. Exact solution $y_t(x) = \cos(x)$, predicted solution $y_p(x)$ by LDNN.

### 4.3   EXPERIMENT 3

Suppose that we have the following model (Babolian et al., 2007):

$$y(x) = g(x) + \int_0^x (x-s)y^2(s)\mathrm{d}s + \int_0^1 (x+s)y(s)\mathrm{d}s, \quad x \in [0,1]. \tag{19}$$

where

$$g(x) = -\frac{1}{30}x^6 + \frac{1}{3}x^4 - x^2 + \frac{5}{3}x - \frac{5}{4} \tag{20}$$

It has the exact solution $y(x) = x^2 - 2$. Table 4 illustrates the exact value, the predicted value and the absolute error (Error) in several test points on $[0,1]$ domain. 50 points of shifted Legendre quadrature points are applied for training LDNN. The number of train data set is 500 and the number of test data set is 100. Figure 4 represented the comparison between $y_t$ and $y_p$.

Table 4: The exact value, the predicted value and the absolute error (Error) in several test points on $[0, 1]$ domain for Experiment 3.

| $x$ | exact value ($y_t = x^2 - 2$) | predicted value ($y_p$) | Error |
|---|---|---|---|
| 0.0 | $-2.0$ | $-2.00000001$ | 9.99999994e$-$09 |
| 0.2 | $-1.96$ | $-1.96000049$ | 4.90000000e$-$07 |
| 0.4 | $-1.84$ | $-1.840000009$ | 8.99999986e$-$09 |
| 0.6 | $-1.64$ | $-1.64000036$ | 3.60000000e$-$07 |
| 0.8 | $-1.36$ | $-1.35999998$ | 2.00000001e$-$08 |
| 1.0 | $-1.0$ | $-0.99999999$ | 1.00000001e$-$08 |

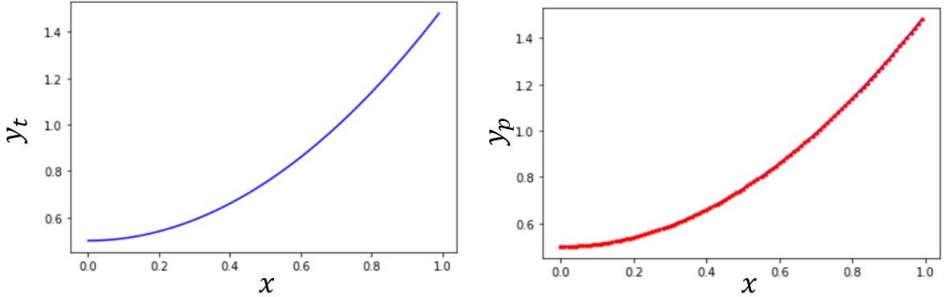

Figure 4: Results of Experiment 3. Exact solution $y_t(x) = x^2 - 2$, predicted solution $y_p(x)$ by LDNN.

### 4.4 EXPERIMENT 4

Suppose that we have the following model (Hadizadeh & Mohamadsohi, 2005):

$$y(x) = -\frac{1}{10}x^4 + \frac{5}{6}x^2 + \frac{3}{8} + \int_0^x \frac{1}{2x}y^2(s)\mathrm{d}s, \quad x \in [0, 1]. \tag{21}$$

It has the exact solution $y(x) = x^2 + \frac{1}{2}$. The exact value, the predicted value and the absolute error (Error) in several test points on $[0, 1]$ domain are reported in Table 5. 50 points of shifted Legendre quadrature points are applied for training LDNN. The number of train data set is 500 and the number of test data set is 100. Figure 5 shows the illustrated comparison between $y_t$ and $y_p$.

Figure 5: Results of Experiment 4. Exact solution $y_t(x) = x^2 + \frac{1}{2}$, predicted solution $y_p(x)$ by LDNN.

### 5 CONCLUSION

Legendre deep neural network (LDNN) is introduced in this paper. LDNN and its application for solving nonlinear Volterra–Fredholm–Hammerstein integral equations (V-F-H-IEs) are proposed.

Table 5: The exact value, the predicted value and the absolute error (Error) in several test points on $[0, 1]$ domain for Experiment 4.

| $x$ | exact value ($y_t = x^2 + \frac{1}{2}$) | predicted value ($y_p$) | Error |
|-----|-----|-----|-----|
| 0.0 | 0.5 | 0.50000004 | 4.00000000e−09 |
| 0.2 | 0.54 | 0.54000001 | 9.99999994e−09 |
| 0.4 | 0.66 | 0.66000002 | 2.00000000e−08 |
| 0.6 | 0.86 | 0.85999998 | 2.00000000e−08 |
| 0.8 | 1.14 | 1.13999999 | 9.99999994e−09 |
| 1.0 | 1.5 | 1.49999999 | 9.99999994e−09 |

LDNN includes two networks. The first network approximates the solution of a nonlinear V-F-H-IE $y(x)$ which has $\mathcal{M}$-layers feed forward neural network structure. The first hidden layer of this has a orthogonal layer consists of Legendre polynomials as activation functions. The last network adjusts the output of the sooner network to fit to a desired equation form. The better performance of the network has been obtained by using Legendre Gaussian integration and automatic differentiation. Some experiments of nonlinear V-F-H-IEs are given to investigate the reliability and validity of LDNN. The results show that this network is an efficient and has high accuracy.

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
