# OpenReview forum: "Legendre Deep Neural Network (LDNN) and its application for approximation of nonlinear Volterra–Fredholm–Hammerstein integral equations"
_ICLR.cc/2021/Conference — Reject_

### Official Review · AnonReviewer3 · 2020-10-23
**Interesting direction, motivation for method must be stronger**

**Rating:** 4
**Confidence:** 4

**Review:**

Overview: The paper proposes to solve the Volterra-Fredholm-Hammerstein integral equations using a form of neural networks: Legendre Deep Neural Networks which use Legendre polynomials in combination with a neural network to model the solution. The idea and direction of the paper is interesting, however both the theory and numerical results could be improved and specifically it should be made more clear *what* is the added benefit of using this LDNN model as opposed to any other method for solving the integral equations. I suggest some ideas for improvements below.

Ideas for improvement, comments and questions:
- In general I like the idea of combining collocation, Legendre polynomials and neural networks.
- For readability (especially for people less familiar with these methods) it would be good to explain some concepts more clearly (see also next points).
- Are the last parameters \zeta_1 and \zeta_2 fixed or trainable? The way it looks in the equation now is since you refer to it as the ‘second network’ that they are trainable, but in Section 3 you mention they are fixed. What are the trainable parameters of the third layer?
- How are the roots $X_j$ and consequently $\omega_j$ computed?
- The numerical results look good and I see the potential of the method.
- What is the intuition behind the FC layers after the Legendre layer? What would more hidden layers mean in terms of the approximation? Some numerical results on this would be useful.
- Related to the above, how does the LDNN method compare to the collocation method and are W^2 similar to the role of the coefficients of the basis functions?
- I think the case of *why* we would be using the LDNN method instead of other more classical approaches should be made a lot stronger for the paper to be of significant contribution. Right now it is not clear for me how the proposed method is better than classical methods. What is the relation to methods which, as you mention, “an attempt is made to obtain the unknown coefficients of the basis functions so that the solution satisfies the equation in a set of candidate points”.
- A small comment but the English of the paper should be improved.
- Maybe introduce Legendre polynomials prior to defining the network?

---

### Official Review · AnonReviewer4 · 2020-10-28
**No improvements in both method and experiments**

**Rating:** 3
**Confidence:** 5

**Review:**

The paper proposed the Legendre Deep Neural Network (LDNN) to solve Volterra–Fredholm–Hammerstein integral equations. Specifically, the network uses Legendre polynomials as the activation in the first layer and uses Gaussian quadrature to discretize the integral operator as a summation. The numerical examples are performed to verify the performance of LDNN. However, the method is not novel and the numerical examples are too simple.

Major comments:

- The proposed method is the same as the physics-informed neural network (PINN) for solving integral PDEs proposed in https://arxiv.org/abs/1907.04502, but this is not mentioned in the paper. In fact, the method proposed in https://arxiv.org/abs/1907.04502 is more general than the method in this paper and can solve more types of PDEs.
- The only difference is that here the first layer of the network uses Legendre polynomials as the activation, but there is not any evidence in the paper that if using Legendre polynomials would make a significant difference.
- The integral equations solved in this paper are very simple. The equations only have one or two integrals, the problem is one dimension, and the solutions are quite smooth. In https://arxiv.org/abs/1907.04502, an integro-differential equation is solved. The same method has also been used to solve the 1D/2D/3D time-fractional/space-fractional/time-space-fractional PDE in a complicated geometry for both forward and inverse problems (https://doi.org/10.1137/18M1229845), which is much harder than the problems solved in this paper.

---

### Official Review · AnonReviewer5 · 2020-11-07

**Rating:** 5
**Confidence:** 2

**Review:**

** Summary **

The authors present a neural network based method to solve a special class of integral equations. Their approach involves training a neural network with Legendre polynomial based activation functions to approximate the solution $y(x)$ for a given $x$. The network is trained in a supervised fashion to minimize a loss function with two term- (1) the $\ell_2$ error between the true solution and $y(x)$ and (2) the residual of the given integral  equation when analysed at $x$. They show impressive numerical results for several instances of VFH-IEs with very low errors. The primary contributions as claimed by the authors are the use of Legendre polynomial based activation functions and creating a differentiable approximation for the integral equation by using Legendre polynomials and Quadrature methods to analyse the integral.

*** Pros ***
1. The numerical results show great efficiency and same to perform at par or better compared to other numerical methods reported in literature.
2. The use of Legendre polynomials as an activation function to approximate the input domain is an interesting method to introduce well understood approximations from the numerical methods community.

*** Cons ***
1. The paper lacks comparisons and ablation studies to show how their model compares to simple supervised training. For example, a simple baseline comparison would be to train a network with similar number of parameters and standard loss functions in a supervised fashion and without the IE residual. This would allow us to analyse the efficacy of the various components of the proposed architecture better.

2. How does the proposed method improve upon traditional numerical methods? I also would like to know the timing comparisons between traditional methods and the proposed neural network method.

3. For Figures 2 and 3, the error between $y_{true}$ and $y_{pred}$ should be plotted as well.

The paper in its current form is not addressing how and why neural networks improve performance over the traditional methods and is also missing relevant comparisons and ablation studies. I will be willing to change my score if the authors add the required experimental results.

---

### Decision · Program_Chairs · 2021-01-07
**Final Decision**

**Decision:**

Reject

**Comment:**

All three reviews for this paper were negative, and the authors did not provide rebuttals or comments on the reviews.  The main drawback of this work identified by the reviewers is that the empirical study is not sufficient (e.g., limited comparisons and ablation studies as well as low-dimensional examples).